# Quantum Scalar-Field Propagator in a Stochastic Gravitational-Plane Wave

**Zbigniew Haba**

Institute of Theoretical Physics, University of Wrocław, Plac Maxa Borna 9, 50-204 Wrocław, Poland;
zbigniew.haba@uwr.edu.pl

**Abstract:** A stochastic metric can appear in classical as well as in quantum gravity. We show that if the linearized stochastic Gaussian gravitational-plane wave has the frequency spectrum $\omega^{4\gamma-1}$ ($0 \leq \gamma < 1$), then the equal-time propagator of the scalar field behaves as $p^{-\frac{1}{1-\gamma}}$ for large momenta. We discuss models of quantum-field theory where such anomalous behavior can appear.

**Keywords:** scalar field propagator; gravitational waves; quantum gravity

## 1. Introduction

The standard model on a stochastic background of gravitational waves or the one supplemented with quantum gravity can be viewed as a theory of matter fields in a random geometry which can be singular at short distances. There are some results [1–7] indicating that gravity can modify the short-distance (equivalently, large-momentum) behavior of matter fields. There are some simplified Lorentz non-invariant models where such a modified short-distance behavior in various directions in space–time is realized [8–10]. The question appears of whether the large-momentum behavior of quantum fields in the background of quantized gravity or in the stochastic background could be experimentally verified, e.g., in high-energy scattering. The detection of gravitational waves [11] raises the questions of whether these waves can be considered as a stream of gravitons and whether their interaction with particles can be treated as a particle-graviton scattering. There is a suggestion [12] that the effect of gravitons can be observed as a noise in the interferometers applied for gravitational-waves detection. In such a model, the quantum equation for geodesic deviation is studied. It is shown that gravitons transform this equation into a stochastic differential equation when changing the particle evolution.

In this paper, we are interested in the problem of how the dynamics of the quantum scalar field can be changed in a linearly polarized stochastic gravitational plane wave moving in the *x*-direction. Such a stochastic metric can arise from stochastic sources of gravitational waves or from particular states of quantum gravity. The stochastic metric corresponding to the plane wave has the spectral function $\rho(\mathbf{k})$ restricted to $k_x \geq 0$ with $k_y = k_z = 0$. We show that if the graviton correlation function is singular at short distances, then the scalar-field propagator decays faster in the momentum space (in comparison to the free-field propagator) in the direction orthogonal to the wave propagation. Such an improved behavior of propagators of matter fields in the background of a quantum gravitational field is important for the renormalization and ultraviolet stability of these theories. The singular correlation functions of the metric (large contribution of high frequencies) emerge as a consequence of the perturbative quantization of gravity. The high-frequency dependence of the gravitational radiation can appear as a result of an inflationary enhancement of the gravitational vacuum contribution [13,14], according to Ref. [15]. The plan of the paper is the following. In Section 2, we discuss the Gaussian stochastic plane waves and their interaction with a quantum scalar field. In Section 3, we consider the quantum states of the linearized gravitational field, which can describe a stochastic metric. In Section 4, the Dyson expansion for the propagator in a simplified



model of a random metric is investigated. In Section 5, we consider the Feynman path integral representing the resummation of the Dyson series. In Section 6, we estimate the behavior of the propagator at large momenta. In Section 7, we summarize our results.

## 2. Quantum Scalar Field in a Stochastic Plane Wave

We consider a perurbation $h_{\mu\nu}(\xi)$ ( $\xi = (t, \mathbf{x})$ ) of the Minkowski metric in the traceless transverse (TT) gauge requiring, in addition, that the metric tensor is diagonal and assuming that $h_{\mu\nu}$ is propagating only along the $x$ axis. Then, the metric is

$$
\begin{aligned}
ds^2 &= g_{\mu\nu}d\xi^\mu d\xi^\nu \equiv dt^2 - (\delta_{jk} + h_{jk})dx^j dx^k \\
&\equiv dt^2 - dx^2 - (1 - h(t,x))dy^2 - (1 + h(t,x))dz^2.
\end{aligned}
\tag{1}
$$

Such a metric can describe a gravitational wave moving along the $x$-axis. It can be considered as a solution of the *TT* conditions $\partial_j h_{jk} = 0$ and $h_j^j = 0$. For the plane wave moving in the positive direction of the $x$-axis [16], $h(t,x) = h(u)$ where $u = t - x$ (the velocity of light $c = 1$).

The propagator of the scalar field (with mass $m$) in a stochastic metric is defined by a functional integral

$$
\begin{aligned}
G(\xi, \xi') &= Z^{-1}\Big\langle \int \mathcal{D}\phi \exp\left(\tfrac{i}{2\hbar}\int d^4x \sqrt{|g|}(g^{\mu\nu}\partial_\mu\phi\partial_\nu\phi - m^2\phi^2)\phi(x)\phi(y)\right)\Big\rangle \\
&= \tfrac{1}{2}i\hbar\Big\langle \mathcal{A}^{-1}(\xi; \xi')\Big\rangle,
\end{aligned}
\tag{2}
$$

where $Z$ is a normalization factor, $g = \det(g_{\mu\nu})$ is the determinant of the metric $g_{\mu\nu}$ and $\mathcal{A}^{-1}(x, y)$ denotes the kernel (Green function) of the inverse of the operator defined by the bilinear form in the Lagrangian of the scalar field in Equation (2)

$$
2\mathcal{A} = |g|^{-\frac{1}{2}}\partial_\mu |g|^{\frac{1}{2}}g^{\mu\nu}\partial_\nu + m^2
\tag{3}
$$

In Equation (2), the metric is understood as an average over a stochastic background of gravitational waves [17] which may have a primordial origin (quantum states of the gravitational field, Section 3) or may come from countless sources emitting gravitational waves in the universe (e.g., from the merging of primordial black holes). The average (2) could also be performed in the Feynman path integral expressing the quantum fluctuations in the metric, as discussed in [18–20]. We assume that $h$ can be approximated by a Gaussian variable. Such an assumption, in the case of a stochastic background, is justified by the central limit theorem of probability theory. If we have small stochastic sources of gravitational radiation then the probability distribution of an infinite sum of stochastic variables (no matter what their individual probability distributions) can be approximated by the Gaussian distribution. In the quantum theory of the next section, we show that in the limit of linearized semiclassical gravity we can use the stochastic Gaussian approximation for quantum plane waves.

## 3. Linearized Quantum Gravity

We consider a model of the quantum scalar field interacting with linearized quantum gravity described by a small perturbation $h_{\mu\nu}$ of the Minkowski metric. In the Hamiltonian framework of this theory, we have the Schrödinger equation [13]

$$
i\hbar\partial_t\psi = (H_{gr}(h) + H_{sc}(h, \phi))\psi,
\tag{4}
$$

where $H_{gr}$ is the Hamiltonian for the linearized gravity (the gravitons) and $H_{sc}$ is the Hamiltonian of the quantum scalar field in an external metric $h$. We discuss the correlation functions

$$
(U_t\psi_0, \phi(\mathbf{x})U_t\phi(\mathbf{x}')\psi_0) = (\psi_t, \phi(\mathbf{x})\phi(t, \mathbf{x}')\psi_t),
\tag{5}
$$

where $U_t$ is the unitary evolution generated by the Hamiltonian (4) and $\phi(t, \mathbf{x}) = U_t \phi(\mathbf{x}) U_t^+$. We can write

$$U_t = \exp(-\frac{i}{\hbar} H_{gr} t) T\left( \exp(-\frac{i}{\hbar} \int_0^t H_{sc}^I(s) ds) \right) \equiv U^{gr}(t) U_{sc}^I(t), \tag{6}$$

where $T(\ldots)$ denotes the time-ordered exponential, $U^{gr}(t) = \exp(-\frac{i}{\hbar} H^{gr} t)$ and

$$H_{sc}^I(s) = U^{gr}(s)^+ H_{sc}(h, \phi) U^{gr}(s) = H_{sc}(h(s), \phi)$$

with

$$h(s, \mathbf{x}) = \exp(\frac{i}{\hbar} H_{gr} s) h(\mathbf{x}) \exp(-\frac{i}{\hbar} H_{gr} s). \tag{7}$$

We assume that the initial state is of the product form $\psi_0(h, \phi) = \psi_0^{gr}(h) \psi_0^{sc}(\phi)$. Then,

$$\begin{aligned} &U_t \psi_0^{gr}(h) \psi_0^{sc}(\phi) \\ &= \left( \exp(-\frac{i}{\hbar} H_{gr} t) \psi_0^{gr}(h) \right) T\left( \exp(-\frac{i}{\hbar} \int_0^t H_{sc}^I(s) ds) \right) \psi_0^{sc}(\phi) \\ &\equiv \psi_t^{gr}(h) \psi_t^{sc}(h(.), \phi), \end{aligned} \tag{8}$$

where $\psi_t^{sc}(h(.), \phi)$ becomes a functional of the quantum field $h(s, \mathbf{x})$. Thus far, the formulas are exact. $h(.)$ on the rhs of Equation (8) means that $H_{sc}(h, \phi) \to H_{sc}(h(s), \phi)$. We are using, in $H_{sc}(h(s), \phi)$, the scalar-field Hamiltonian with an operator metric $h(s, \mathbf{x})$ (7). We make the approximation neglecting the non-commutativity of $h(s, \mathbf{x})$ at different times. $h(s, \mathbf{x})$ becomes a classical stochastic field. The commutator $[h(s, \mathbf{x}), h(s', \mathbf{x}')] \simeq O(\hbar)$. This means that neglecting non-commutativity, we are applying a semiclassical approximation to the coupling of the metric to the quantum scalar field. We choose as $\psi_0^{sc}(\phi)$ the ground state of the massive free scalar field. We apply the functional representation of states in quantum-field theory. In this representation, the propagator in a stochastic gravitational field of the quantum scalar field in its ground state is

$$\begin{aligned} &(U_t \psi_0, \phi(\mathbf{x}) \phi(t, \mathbf{x}') U_t \psi_0) \\ &= \int \mathcal{D} h_{rl} |\psi_t^{gr}|^2 \mathcal{D} \phi \exp\left( \frac{i}{2\hbar} \int d^4 x \sqrt{|g|} (g^{\mu\nu} \partial_\mu \phi \partial_\nu \phi - m^2 \phi^2) \right) \phi(\mathbf{x}) \phi(t, \mathbf{x}') \\ &= \frac{1}{2} i\hbar \int \mathcal{D} h_{rl} |\psi_t^{gr}(h)|^2 \mathcal{A}^{-1}(0, \mathbf{x}; t, \mathbf{x}'). \end{aligned} \tag{9}$$

The Hamiltonian for the linearized (free) gravitational field is [13]

$$H_{gr} = \frac{1}{2} \int d\mathbf{k} \left( -\Lambda \frac{\delta}{\delta h_{rl}(\mathbf{k})} \Lambda \frac{\delta}{\delta h_{rl}(-\mathbf{k})} + k^2 (\Lambda h)_{rl}(\mathbf{k})(\Lambda h)_{rl}(-\mathbf{k}) \right), \tag{10}$$

where $r, l = 1, 2, 3$ and $(\Lambda T)_{ij}$ is the projection of a tensor $T_{mn}$ onto the one in the TT gauge defined at the beginning of Section 2. The matrix $\Lambda$ in the momentum representation has the form

$$\begin{aligned} 2\Lambda_{ij;mn}(\tfrac{\mathbf{k}}{k}) &= (\delta_{im} - k^{-2} k_i k_m)(\delta_{jn} - k^{-2} k_j k_n) \\ &+ (\delta_{in} - k^{-2} k_i k_n)(\delta_{jm} - k^{-2} k_j k_m) - \tfrac{2}{3} (\delta_{ij} - k^{-2} k_i k_j)(\delta_{nm} - k^{-2} k_n k_m). \end{aligned}$$

It follows from Equations (7) and (10) that

$$(\partial_t^2 - \triangle) h_{rl} = 0. \tag{11}$$

Moreover, as shown in [13], the correlation functions in the ground state of the Hamiltonian (10) (as calculated below in Equation (17)) satisfy the Weinberg requirements for massless tensor fields [21].

We choose as an initial state the general Gaussian translation invariant wave function of tensorial fields in the TT gauge

$$\psi_0^{gr} = A_0 \exp\left(-\frac{1}{2\hbar}\int(\Lambda h)_{rl}(\mathbf{x})\Gamma_0(\mathbf{x}-\mathbf{y})_{rl,ij}(\Lambda h)_{ij}(\mathbf{y})d\mathbf{x}d\mathbf{y}\right), \tag{12}$$

where $A_0$ is a normalization constant. The Gaussian states have a positively definite Wigner function. For this reason, they give a proper semiclassical approximation for quantum states. For a free Hamiltonian (10), the time evolution of a Gaussian state is again a Gaussian state. The Schrödinger equation $i\hbar\partial_t\psi = H_{gr}\psi$ with the initial condition (12) has the solution which is again a Gaussian translation invariant wave function

$$\psi_t^{gr} = A_t \exp\left(-\frac{1}{2\hbar}\int(\Lambda h)_{rl}(\mathbf{x})\Gamma_t(\mathbf{x}-\mathbf{y})_{rl,ij}(\Lambda h)_{ij}(\mathbf{y})d\mathbf{x}d\mathbf{y}\right), \tag{13}$$

if $\Gamma$ satisfies the equation

$$i\partial_t\Gamma_t(\mathbf{k})_{rl,ij} - \Gamma_t(\mathbf{k})_{rl,mn}\Gamma_t(-\mathbf{k})_{mn,ij} + k^2\delta_{ri}\delta_{lj} = 0, \tag{14}$$

where $\Gamma(\mathbf{k})$ is the Fourier transform of $\Gamma(\mathbf{x})$ and $k = |\mathbf{k}|$. The time-independent solution of Equation (14)

$$\Gamma_t(\mathbf{k})_{rl,ij} = k\delta_{ri}\delta_{lj} \tag{15}$$

corresponds to the ground state [13,14]

$$\psi_0^{gr}(h) = \exp\left(-\frac{1}{2}\int d\mathbf{k}|\mathbf{k}|(\Lambda h)_{jl}(\mathbf{k})^*(\Lambda h)_{jl}(\mathbf{k})\right)$$

leading to the Gaussian measure

$$d\mu_0(h) = \mathcal{D}h_{jk}|\psi_0^{gr}|^2$$

describing in the TT gauge the free tensorial massless fields in the ground state. Then, correlation functions in the ground state are

$$(\psi_0^{gr}, h_{jl}(\mathbf{x})h_{mn}(t,\mathbf{x}')\psi_0^{gr}) = (2\pi)^{-3}\int d\mathbf{k}|\mathbf{k}|^{-1}\exp(-ikt)\exp(i\mathbf{k}(\mathbf{x}-\mathbf{x}'))$$
$$(\delta_{jm}(\mathbf{k})\delta_{ln}(\mathbf{k}) + \delta_{jn}(\mathbf{k})\delta_{lm}(\mathbf{k}) - \delta_{jl}(\mathbf{k})\delta_{mn}(\mathbf{k})), \tag{16}$$

where

$$\delta_{jl}(\mathbf{k}) = \delta_{jl} - k_jk_l|\mathbf{k}|^{-2}.$$

The correlation (16) behaves as

$$\left((\xi-\xi')^2\right)^{-1} = \left((u-u')(x-t-x') - (y-y')^2 - (z-z')^2\right)^{-1} \tag{17}$$

at short space-time distances (in the notation of Section 2).

The time-dependent solutions of Equation (14) can be related to solutions of a linear equation. Let us define the matrix $v(\mathbf{k})$ as a time-ordered exponential of the matrix $\Gamma(\mathbf{k})$

$$v = T\left(\exp(i\int_0^t\Gamma_s ds)\right). \tag{18}$$

It can also be defined as the solution of the equation $\partial_t v = i\Gamma v$. If $\Gamma$ satisfies Equation (14) then the matrix $v$ satisfies a linear equation

$$\frac{d^2v}{dt^2} + k^2v = 0. \tag{19}$$

We can recover $\Gamma$ from $v$ as

$$i\Gamma_t = v^{-1}\frac{dv}{dt}. \tag{20}$$

The general solution of Equation (19) is

$$v = M\exp(ikt) + N\exp(-ikt), \tag{21}$$

where the matrices $M$ and $N$ may depend on $\mathbf{k}$. We have from Equations (20) and (21) that

$$\Gamma_0 = k(M - N)(M + N)^{-1} \tag{22}$$

and

$$\Gamma_t(\mathbf{k}) = k(\exp(ikt)M - N\exp(-ikt))(\exp(ikt)M + N\exp(-ikt))^{-1}. \tag{23}$$

The two-point correlation function of $h$ calculated as the covariance of the measure $d\mu = \mathcal{D}h_{rl}|\psi_t^{gr}|^2$ is the Fourier transform of $\Gamma_t^{-1}(\mathbf{k})$. By a proper choice of the (matrix) functions $M$ and $N$ we can achieve that the dependence of matrix elements of $\Gamma^{-1}$ on the components of $\mathbf{k}$ can be dominating or negligible. We can choose $\Gamma$ in Equations (22) and (23) in such a way that only the correlation functions $< h_{33}h_{33} > = < h_{22}h_{22} > = - < h_{22}h_{33} >$ are not negligible (we demand $(h_{33} = -h_{22} = h$ in the sense of correlation functions). The remaining components of the metric in the TT gauge $h_{11}$ and $h_{jk}$ for $j \neq k$ will have vanishing correlation functions by a choice of $\Gamma$. We choose $\Gamma_{rl,ij}$ such that $\Gamma_{rl,rl} \to \infty$ for $rl = 11, 12, 13, 23$. In such a case the expectation values (and correlation functions) with respect to the measure $d\mu = \mathcal{D}h_{rl}|\psi_t^{gr}|^2$ of the components of the tensor $h_{rl}$ which are absent in the metric (1) tend to zero. In this way the wave function $\psi_t^{gr}$ is imposing the TT gauge in the sense that the correlation functions of $h$ asymptotically satisfy this gauge. The TT gauge (for diagonal $h$) has as a consequence that $h$ depends only on $t$ and $x$. It follows that the wave Equation (11) takes the form $(\partial_t^2 - \partial_x^2)h = 0$ (in the sense of expectation values) with the solution which is a sum of waves depending either on $t - x$ or on $t + x$. We can damp the $t + x$ component choosing $\Gamma_{22,22}(\mathbf{k}) = \Gamma_{33,33}(\mathbf{k}) \to \infty$ when $k_x < 0$.

Summarizing, we have shown in this section that a gravitational wave treated as a metric in a particular quantum state of a linearized quantum gravity can be considered as a stochastic Gaussian plane wave. We discuss a scalar quantum field in such a stochastic wave in subsequent sections.

### 4. Dyson Expansion of the Scalar Field Propagator

We consider the metric (1). We assume that we have a small perturbation of the Minkowski metric, so that $h^2 \simeq 0$. In such a case the term $\partial_t \ln|g|\partial_t - \partial_x \ln|g|\partial_x$ which is of the first order in derivatives for the scalar wave operator in Equation (3) is absent because $g = \det(g_{jk}) = 1 - h^2 \simeq 1$. For the same reason we make the approximation $(1 \pm h)^{-1} \simeq 1 \mp h$. If $h$ depends only on $u = t - x$ then we can change the variable $t \to t - x$ in $\mathcal{A}$. Then, the coefficients of the differential operator $\mathcal{A}$ in Equation (3) depend on one variable $u$

$$\mathcal{A} = -\frac{1}{2}(\partial_u^2 - \triangle + m^2) + \frac{1}{2}h(u)\partial_y^2 - \frac{1}{2}h(u)\partial_z^2. \tag{24}$$

We can take the Fourier transform in spatial variables of the operator (24) and its kernel

$$\tilde{\mathcal{A}}^{-1}(u, u', \mathbf{p}) = \int d\mathbf{x}\exp(ip_x(x - x') + ip_y(y - y') + ip_z(z - z'))\mathcal{A}^{-1}(\xi; \xi') \tag{25}$$

It follows that $\tilde{\mathcal{A}}^{-1}(u, u', \mathbf{p})$ is the kernel of the operator

$$\tilde{\mathcal{A}} = -\frac{1}{2}(\partial_u^2 + \mathbf{p}^2 + m^2) + \frac{1}{2}h(u)p_+p_- \equiv \tilde{\mathcal{A}}_0 + V \tag{26}$$



acting in the space of functions $\psi(u, \mathbf{p})$ (where the action of the momentum $\mathbf{p}$ in $\tilde{\mathcal{A}}$ is just the multiplication). Here, $\mathbf{p}^2 = p_x^2 + p_y^2 + p_z^2$, $p_+ = p_z + p_y$, $p_- = p_z - p_y$,

$$\tilde{\mathcal{A}}_0 = -\frac{1}{2}\partial_u^2 - \frac{1}{2}\mathbf{p}^2 - \frac{1}{2}m^2 \tag{27}$$

and

$$V = \frac{1}{2}h(u)p_-p_+. \tag{28}$$

With minor changes in the discussion of the propagator (25) we can consider the case of the metric $h$ depending only on $x$. This changes $\tilde{\mathcal{A}}_0$ in Equation (26) as

$$\tilde{\mathcal{A}}_0 = \frac{1}{2}\partial_x^2 + \frac{1}{2}p_0^2 - \frac{1}{2}\mathbf{p}_\perp^2 - \frac{1}{2}m^2, \tag{29}$$

where

$$\mathbf{p}_\perp^2 = p_y^2 + p_z^2. \tag{30}$$

The form of the operator $V$ (28) does not change except that $h$ depends only on $x$.

We shall consider a random metric $h$. This randomness can come either from classical sources as discussed in Section 2 or from quantum fluctuations of Section 3. For explicit calculations we need a dependence of $h$ solely on one coordinate. There can be a physical reason for such an approximation. In the case of the classical waves this requirement relies on the assumption that we deal with plane waves. In the quantum realm such a metric depending solely on $t - x$ or on $x$ can arise from a particular initial state as discussed in Section 3.

We perform the calculations for the operator $\tilde{\mathcal{A}}$ (26) with $\tilde{\mathcal{A}}_0$ (27) corresponding to the plane wave. The calculations with $\tilde{\mathcal{A}}_0$ (29) are similar (just replace $\tilde{\mathcal{A}}$ by $-\tilde{\mathcal{A}}$ and $u$ by $x$). We represent the propagator $\tilde{\mathcal{A}}^{-1} = i \int_0^\infty d\tau \exp(-i\tau\tilde{\mathcal{A}})$ in terms of the proper time Hamiltonian evolution. The operator $\tilde{\mathcal{A}}$ has the form of a Hamiltonian in quantum mechanics. We assume that it has a self-adjoint extension. The unitary evolution $\exp(-i\tilde{\mathcal{A}}\tau)$ can be expressed in the interaction picture. We write

$$U_\tau = \exp(-i\tilde{\mathcal{A}}\tau) = \exp(-i\tilde{\mathcal{A}}_0\tau)U_\tau^I. \tag{31}$$

Then,

$$\partial_\tau U_\tau^I = -iV_\tau U_\tau^I, \tag{32}$$

where

$$V_\tau = \exp(i\tilde{\mathcal{A}}_0\tau)V\exp(-i\tilde{\mathcal{A}}_0\tau). \tag{33}$$

We are interested in an average over the random field $h$. We can calculate such averages expanding the solution of Equation (32) in the Dyson series

$$U_\tau = \exp(-i\tilde{\mathcal{A}}_0\tau)U_\tau^I = \exp(-i\tilde{\mathcal{A}}_0\tau)\left(1 - i\int_0^\tau dsV_s - \int_0^\tau ds_2 \int_0^{s_2} ds_1 V_{s_2}V_{s_1} + ...\right). \tag{34}$$

The expectation value over the metric in the lowest order is

$$\begin{aligned}
&< (U_\tau\psi)(u) >= (\exp(-i\tau H_0)\psi)(u)\\
&- \int_0^\tau ds_2 \int_{-\infty}^\infty du_1 \int_{-\infty}^\infty du_2 (\exp(-i(\tau - s_2)H_0))(u, u_1) < V(u_2)V(u_1) >\\
&\times(\exp(-i(s_2 - s_1)H_0))(u_1, u_2) \int_0^{s_2} ds_1 (\exp(-is_1 H_0))(u_2, u_3)\psi(u_3)du_3,
\end{aligned} \tag{35}$$

where

$$\begin{aligned}
&(\exp(-is\tilde{\mathcal{A}}_0))(u, u') = (2i\pi s)^{-\frac{1}{2}} \exp\left(\frac{i}{2s}|u - u'|^2\right) \exp(i\frac{s}{2}(m^2 + \mathbf{p}^2))\\
&\equiv \exp(i\frac{s}{2}(m^2 + \mathbf{p}^2))p(s; u - u').
\end{aligned} \tag{36}$$

Going from Equation (34) to Equation (35) we have changed the proper time integration variables $s_1 \to \tau - s_2$ and $s_2 \to \tau - s_1$.

## 5. Feynman Path Integral Representation

For a non-perturbative averaging over $h$ it is useful to express the evolution $U_\tau$ by means of the Feynman path integral

$$
\begin{aligned}
(U_\tau \psi)(u) &= \exp(i\tau \tfrac{1}{2}(\mathbf{p}^2 + m^2)) \int_{q(0)=u} \mathcal{D}q(.) \\
&\times \exp\left(\tfrac{i}{2} \int_0^\tau ds (\tfrac{dq}{ds})^2\right) \exp(-i \int_0^\tau ds V(q(s))) \psi(q(\tau)).
\end{aligned}
\tag{37}
$$

The mean value of the evolution operator in the Gaussian random metric can be evaluated as

$$
\begin{aligned}
< (U_\tau \psi)(u) > &= \exp(i\tau \tfrac{1}{2}(\mathbf{p}^2 + m^2)) \int_{q(0)=u} \mathcal{D}q(.) \exp\left(\tfrac{i}{2} \int_0^\tau (\tfrac{dq}{ds})^2\right) \\
&\exp(-\tfrac{1}{2} \int_0^\tau ds \int_0^\tau ds' < V(q(s))V(q(s')) >) \psi(q(\tau)).
\end{aligned}
\tag{38}
$$

The Feynman path integral (37) can be considered as a resummation of the Dyson series (34). The lowest order term in the expansion of the Feynman formula for $U_\tau \psi$ in Equation (37) is

$$
\begin{aligned}
< (U_\tau \psi)(u) > &= \exp(i\tau \tfrac{1}{2}(\mathbf{p}^2 + m^2)) \int_{q(0)=u} \mathcal{D}q(.) \exp\left(\tfrac{i}{2} \int_0^\tau (\tfrac{dq}{ds})^2\right) \\
&\times \left(1 - \int_0^\tau ds_2 \int_0^{s_2} ds_1 < V_{s_2} V_{s_1} > + ...\right) \psi(q(\tau)).
\end{aligned}
\tag{39}
$$

It agrees with the Dyson expansion (35). The equality of the expansion in $V$ of the Feynman integral (38) and (39) and the Dyson expansion (34) and (35) can be shown using the expression for the calculation of the Feynman integral of "cylinder functions" [22,23]. If $0 \le s_1 \le s_2 \le ..... \le s_{n-1} \le \tau$ then the Feynman integral of functions of paths starting from $q(0) = u$ and depending on a finite set of points (cylinder functions) $(q(s_1), q(s_2), ...., q(s_{n-1}), q(\tau))$ is

$$
\begin{aligned}
&\int_{q(0)=u} \mathcal{D}q(.) \exp\left(\tfrac{i}{2} \int_0^\tau (\tfrac{dq}{ds})^2\right) F(q(s_1), q(s_2), ...., q(s_{n-1}), q(\tau)) \\
&= \int p(s_1, u_1 - u) p(s_2 - s_1; u_2 - u_1) .... p(\tau - s_{n-1}; u_n - u_{n-1}) \\
&\times F(u_1, ..., u_n) du_1 ... du_n
\end{aligned}
\tag{40}
$$

for any function $F(u_1, ..., u_n)$ of $n$-variables. Equations (38) and (39) coincide with Equation (35) if in the $(n-1)$th order of Dyson perturbation expansion we change the proper time integration variables $s_j \to \tau - s_{n-j}$ (as we did in Equation (35) for $n = 3$).

We may express the time evolution in terms of the evolution kernel $K_\tau$

$$
(U_\tau \psi)(u) = \int du' K_\tau(u, u'; \mathbf{p}) \psi(u').
\tag{41}
$$

The two-point function for the scalar field is obtained as the resolvent of $\tilde{A}$

$$
G(\tilde{A}) = i \int_0^\infty d\tau \exp(-i\tilde{A}\tau) = \tilde{A}^{-1}.
$$

Its kernel is (in time coordinates $u$ and in spatial momenta)

$$
G(\tilde{A}, u, u'; \mathbf{p}) = i \int_0^\infty d\tau \exp(-i\tilde{A}\tau)(u, u') = i \int_0^\infty d\tau K_\tau(u, u'; \mathbf{p}).
\tag{42}
$$

As discussed in Sections 2 and 3, we impose on the Gaussian measure $d\mu(h) = \mathcal{D}h_{rl} |\psi_t^{gr}|^2$ the conditions that in the linearized quantum gravity the measure is strongly concentrated on the metric solving the wave equation and satisfying the TT condition. In such a case (as discussed at the end of Section 3), the metric depends only on $u = t - x$. The background of plane gravitational waves, discussed in Section 2, moving in a fixed direction can also

depend on one variable. In these idealized cases, we assume that $h(u)$ is a Gaussian random field defined by the measure $d\mu(h)$ characterized by its generating functional

$$\int d\mu(h) \exp\left(i \int du J(u)h(u)\right) = \exp\left(-\frac{1}{2}\int du du' J(u)g(u-u')J(u')\right), \quad (43)$$

where $g(u-u')$ is the correlation function of $h(u)$. We consider the covariance

$$g(u-u') = < h(u)h(u') > = \kappa^2|u-u'|^{-4\gamma}. \quad (44)$$

We need $0 < \gamma < \frac{1}{4}$ for a rigorous approach (the restriction $\gamma < \frac{1}{4}$ and consequences of crossing the line $\gamma = \frac{1}{4}$ are considered at the end of Section 6). The exact scale invariance of $g(u)$ is not necessary for our results. It is sufficient to assume that $g(u)$ has the form (44) for $u \to 0$. We make the assumption of an exact scale invariance for simplicity of the argument. The absolute value $|u-u'|$ in Equation (44) is needed for a mathematical definition of a stochastic variable, as the bilinear form in the exponential in Equation (44) is positively definite only with the absolute value and $0 < \gamma < \frac{1}{4}$. However, in the next section, only the scaling property of the covariance (44) is used in a derivation of the result. Let us note that the free-field correlation function (17) behaves as $(u-u')^{-1}$ (so $\gamma = \frac{1}{4}$) if $y = y' = z = z' = 0$). In the free field correlation (17) there is no absolute value but instead the $i\epsilon$ prescription for an interpretation of the integral over $u, u'$ in the Feynman propagator. We set $\gamma$ as a free parameter taking into account the suggestion [1–4,6] that quantum gravity at short distances can be more regular than in canonical field theory. For a stochastic background of gravitational waves, the scale invariant correlation functions (44) were derived in some models of gravitational radiation [24,25]. It is our aim in this paper to show that singular gravity leads to a more regular scalar field propagator. The singularity of the correlations at small time means a large contribution of high-frequency waves to the spectrum (as the frequency spectrum of the covariance (44) is $\omega^{4\gamma-1}$). At present, only low-frequency gravitational waves have been detected [17]. The large contribution of high-frequency modes to the wave spectrum is expected to have a quantum origin. These quantum fluctuations could be enhanced by inflation and appear in an observable wave spectrum, as discussed in [15] (a similar mechanism is known to work for scalar fields leading to squeezed states [26,27]).

## 6. Estimates on the Propagator

We have for a Gaussian field $h(u)$

$$< G(\check{A}, u, u'; \mathbf{p}) >= i \int_0^\infty d\tau \exp(i\tau \frac{1}{2}(\mathbf{p}^2 + m^2)) \int_{q(0)=u} \mathcal{D}q(.) \exp\left(\frac{i}{2}\int_0^\tau (\frac{dq}{ds})^2\right)$$
$$\times \exp(-\frac{1}{2}\int_0^\tau ds \int_0^\tau ds' < V(q(s))V(q(s')) >)\delta(u' - q(\tau)), \quad (45)$$

so that the path $q(s)$ starts at $u$ and ends in $u'$.

Let us note the identity for $q(s)$ (in the sense of the equality of expectation values; a consequence of the transition function $p(s; u)$ in Equation (36))

$$q(s) = \sqrt{\tau}q(\frac{s}{\tau}). \quad (46)$$

Then, in Equation (45) with the correlations (44), we have

$$< G(\check{A}, u, u'; \mathbf{p}) >= i \int_0^\infty d\tau \exp(i\tau \frac{1}{2}(\mathbf{p}^2 + m^2)) \int_{q(0)=\sqrt{\tau}u} \mathcal{D}q(.) \exp\left(\frac{i}{2}\int_0^1 (\frac{dq}{ds})^2\right)$$
$$\exp(-\frac{1}{2}\tau^{2(1-\gamma)}(p_+p_-)^2\kappa^2 \int_0^1 ds \int_0^1 ds' < g(\mathbf{q}(s) - \mathbf{q}(s')) > \delta(u' - \sqrt{\tau}\mathbf{q}(1)). \quad (47)$$

The kernel $K_\tau(u, u'; \mathbf{p})$ (41) can be expressed in terms of the Brownian bridge $Q$ [23] (its Feynman version is derived in [28]), defined as the Gaussian process on the interval $[0, 1]$ ($Q(0) = 0$ and $Q(1) = 0$) with the covariance ($s, s' \geq 0$)

$$< Q(s)Q(s') > = is(1 - s')\theta(s' - s) + is'(1 - s)\theta(s - s'), \tag{48}$$

where $\theta$ is the Heaviside step function. Let us denote by $d\nu(Q)$ the Gaussian measure with the covariance (48). Then,

$$\begin{aligned}
K_\tau(u, u'; \mathbf{p}) &= (2i\pi\tau)^{-\frac{1}{2}} \exp\left(\tfrac{i}{2\tau}|u - u'|^2\right) \exp(\tfrac{i\tau}{2}(\mathbf{p}^2 + m^2)) \\
&\int d\nu(Q) \exp\left(i\tau p_+ p_- \int_0^1 ds h(u's + (1 - s)u + \sqrt{\tau}Q(s))\right)
\end{aligned} \tag{49}$$

The expectation value of the propagator (47) is

$$\begin{aligned}
< G(\tilde{\mathcal{A}}; u, u'; \mathbf{p}) > &= i \int_0^\infty d\tau (2i\pi\tau)^{-\frac{1}{2}} \exp\left(\tfrac{i}{2\tau}|u - u'|^2\right) \\
&\exp(i\tau\tfrac{1}{2}(\mathbf{p}^2 + m^2)) \int d\nu(Q) \exp\left(-\tfrac{1}{2}\tau^{2(1-\gamma)}(p_- p_+)^2 \kappa^2 \int_0^1 ds \int_0^1 ds'\right. \\
&\left.\times g(\tfrac{1}{\sqrt{\tau}}(u' - u)(s - s') + Q(s) - Q(s'))\right).
\end{aligned} \tag{50}$$

In Equations (47) and (50) we applied the scaling property of $g(u)$ (44).

At the end of this section, let us return to the expansion (35) (or (39)) and calculate it with the correlation function (44)

$$\begin{aligned}
&\int_{q(0)=u} \mathcal{D}q(.) \exp\left(\tfrac{i}{2}\int_0^\tau (\tfrac{dq}{ds})^2\right) \int_0^\tau ds_2 \int_0^{s_2} ds_1 < V_{s_2}V_{s_1} > + ...\right)\psi(q(\tau)) \\
&= \kappa^2 (p_+ p_-)^2 \int_0^\tau ds_2 \int_0^{s_2} ds_1 p(s_1; u - u_1)p(s_2 - s_1; u_2 - u_1) \\
&\times p(\tau - s_2; u_3 - u_2)|u_2 - u_1|^{-4\gamma}\psi(u_3)du_3 + ....
\end{aligned} \tag{51}$$

The integral over $u_2 - u_1$ exists as the Lebesgue integral only if $\gamma < \frac{1}{4}$ (for $\gamma = \frac{1}{4}$ we have a logarithmic divergence). We encounter the same problem when calculating (perturbatively) the expectation value of the exponential in Equation (47). Such integrals could possibly be interpreted in the sense of generalized functions [29], allowing an extension to $\gamma > \frac{1}{4}$ (the ground-state correlation function (17) corresponds to $\gamma = \frac{1}{4}$). Such an extension could allow the definition of the s-integrals in Equations (50) and (51). However, some positivity properties of the integrals may be lost, which can lead to difficulties in the interpretation of the result in the framework of quantum-field theory.

An explicit calculation of the propagator (50) can be performed in a perturbation expansion in $\kappa$. The result is equivalent to the calculation by means of the Dyson series (35). When we set $u = u'$, then we can obtain from Equation (50) some non-perturbative estimates. Let us change the proper time variable to $\tau = \tau'(\kappa p_+ p_-)^{-\frac{1}{(1-\gamma)}}$. Then, at $u = u'$ in the exponential in Equation (50), if $0 < \gamma < 1$ the term $\mathbf{p}^2\tau = \mathbf{p}^2\tau'(\kappa p_+ p_-)^{-\frac{1}{(1-\gamma)}}$ becomes small in comparison to the term $\tau'^{2(1-\gamma)}$ in the limit of large $p_z$ (or $p_y$). Performing the $\tau'$ integral (with the negligence of the $\mathbf{p}^2\tau$ term), we obtain

$$< G(\tilde{\mathcal{A}}; u, u; \mathbf{p}) > = C(p_+ p_-)^{-\frac{1}{2(1-\gamma)}} \tag{52}$$

with a certain constant $C$ (this constant can be an infinite renormalization constant if $\gamma \geq \frac{1}{4}$ because of the divergence of the integral (51)). If there is no stochastic metric ($\kappa = \gamma = 0$ in Equation (50)), then by means of the integration over $\tau$ we obtain the propagator of the scalar field at equal time $u = u'$ as

$$(\mathbf{p}^2)^{-\frac{1}{2}}. \tag{53}$$

We can see that if $1 > \gamma > 0$ then the propagator (52) is decaying faster for large $p_z$ than the free propagator (53).

Although the propagator in a stochastic metric (44) is not Lorentz invariant, it is instructive to calculate the Lorentz invariant propagators with a non-canonical scaling, as discussed in [30,31]

$$
\begin{aligned}
G(\xi) &= i \int dp_0 d\mathbf{p} \exp(i\xi^\mu p_\mu) \int_0^\infty d\tau (i\tau)^\alpha \exp(i\tfrac{\tau}{2}p^2) \\
&= (2\pi)^{-2} \int_0^\infty d\tau (i\tau)^{-2+\alpha} \exp(\tfrac{i}{2\tau}\xi^2)
\end{aligned}
\tag{54}
$$

where $p^2 = p_0^2 - \mathbf{p}^2$ and $\xi^2 = t^2 - \mathbf{x}^2$. From Equation (54), the propagator in the momentum space is $(p^2)^{-1-\alpha}$ and the propagator at $t = 0$ is $(\mathbf{p}^2)^{-\frac{1}{2}-\alpha}$. Hence, in order to obtain the behavior (52) of the equal time propagator (50) for a large $p_z$, we need $(1 - \gamma)^{-1} = 1 + 2\alpha$.

## 7. Summary

We derived an asymptotic behavior in the momentum space of the scalar-field two-point correlation function (the propagator) in particular Gaussian Lorentz non-invariant states. The states describe a plane wave moving in the $x$ direction whose metric has singular short-distance correlation functions. As a result of an interaction of the scalar field with this quantum fluctuation in the metric, the scalar-field propagator decays faster in the momentum space (in direction orthogonal to the wave motion) than is possible in canonical field theory. Canonical field theory allows the faster decay of correlations at large distances (small momenta) but not at small distances (large momenta). From the method of derivation of the result, it can be seen that such an anomalous behavior cannot arise from a coupling of the scalar field to a quantum gauge field or to another scalar field, but is characteristic of the coupling of the metric to the kinetic part of the scalar-field Lagrangian. Another special feature of the model is that we did not calculate the scalar-field correlations in the ground state of the metric field but in a particular time-dependent Lorentz non-invariant solution of the Schrödinger equation. It is possible that in a Lorentz invariant quantum theory of matter fields interacting with quantum gravity the scalar field expectation values in a time-dependent states can decay faster in the momentum space than this is allowed in vacuum states (if such states exist at all in theories with quantum gravity). In this way, there remains the prospect of a construction of quantum-field theory including gravity where matter-field correlation functions in time-dependent states are more regular at short distances than is allowed in the vacuum states of Lorentz invariant theory (according to the Källen-Lehmann representation). Such a scheme would be a realization of the regularizing role of Wheeler's "quantum foam", as reviewed recently in [7].

**Funding:** This research received no external funding.

**Data Availability Statement:** Not applicable.

**Conflicts of Interest:** The author declares no conflict of interest.

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
