# Peer review of "Quantum Scalar-Field Propagator in a Stochastic Gravitational-Plane Wave"

_universe, doi:10.3390/universe8120648_

Round 1
Reviewer 1 Report
Referee review for Quantum scalar field propagator in a stochastic gravitational plane wave
by Zbigniew Haba
In this work the author studies the behaviour of quantum fields under a stochastic metric. The properties of plane waves, the corresponding
linearized quantum gravity, the Dyson expansion of the scalar field propagator and Feynman representation are well presented. Many derivations are computed
under relevant physical assumptions. Moreover, the behaviour of the propagator for large momentum is found not to follow the asymptotic behavior of the Kaellen-
Lehmann representation of the vacuum correlation functions of the scalar field in QFT.
I found the article well presented and well explained with the most important relations derived. I did not find any flaw to this study, therefore I would like to recommend it for publication.
Author Response
Thank you
Author Response
I have replied to referee in the file refuni.pdf
